# Apremilast as a Potential Targeted Therapy for Metabolic Syndrome in Patients with Psoriasis: An Observational Analysis

**DOI:** 10.3390/ph17080989

**Published:** 2024-07-26

**Authors:** Elena Campione, Nikkia Zarabian, Terenzio Cosio, Cristiana Borselli, Fabio Artosi, Riccardo Cont, Roberto Sorge, Ruslana Gaeta Shumak, Gaetana Costanza, Antonia Rivieccio, Roberta Gaziano, Luca Bianchi

**Affiliations:** 1Dermatology Unit, Department of Systems Medicine, Tor Vergata University Hospital, 00133 Rome, Italy; crisborselli@gmail.com (C.B.); fabio.artosi994@gmail.com (F.A.); riccardo.cont@hotmail.it (R.C.); ruslanagaetashumak@gmail.com (R.G.S.); antonia.rivieccio@libero.it (A.R.); luca.bianchi@uniroma2.it (L.B.); 2School of Medicine and Health Sciences, George Washington University, 2300 I St NW, Washington, DC 20052, USA; nzarabian@gwmail.gwu.edu; 3Department of Experimental Medicine, University of Rome Tor Vergata, Via Montpellier 1, 00133 Rome, Italy; terenziocosio@gmail.com (T.C.); roberta.gaziano@uniroma2.it (R.G.); 4Dynamyc Research Team 7380, Université de Paris-Est-Créteil, 94000 Créteil, France; 5Laboratory of Biometry, Department of Systems Medicine, University of Rome Tor Vergata, 00133 Rome, Italy; sorge@uniroma2.it; 6Unit of Virology, Department of Experimental Medicine, University of Rome Tor Vergata, Via Montpellier 1, 00133, Rome, Italy; costanza@med.uniroma2.it

**Keywords:** apremilast, psoriasis, comorbidities, metabolic syndrome, PDE4-inhibitor

## Abstract

Psoriasis (PsO) is a chronic inflammatory dermatosis that often presents with erythematous, sharply demarcated lesions. Although psoriasis is primarily a dermatological disease, its immune-mediated pathogenesis produces systemic effects and is closely associated with various comorbid conditions such as cardiovascular disease (CVD), metabolic syndrome (MetS), and diabetes mellitus type II (DMII). Apremilast, an oral phosphodiesterase 4 (PDE-4) inhibitor, has shown promise in treating moderate-to-severe psoriasis and is associated with potential cardiometabolic benefits. In a 12-month prospective observational study involving 137 patients with moderate-to-severe psoriasis, we assessed changes in psoriasis clinimetric scores and metabolic profiles from baseline (T0) to 52 weeks (T1) to evaluate the efficacy of apremilast. After 52 weeks of apremilast treatment, we documented a statistically significant decrease in low-density lipoprotein (LDL) and total cholesterol, triglycerides, and glucose levels. Our findings even suggest a potential synergistic effect among patients treated with apremilast, alongside concomitant statin and/or insulin therapy. Although the results of our study must be validated on a larger scale, the use of apremilast in the treatment of psoriatic patients with cardio-metabolic comorbidities yields promising results.

## 1. Introduction

Psoriasis is a chronic inflammatory dermatosis that affects over 60 million adults and children [1] or approximately 2–3% of the world’s population [2]. This immune-mediated disease is caused by an overlap of genetic and environmental risk factors and often appears in the second-to-fourth decade of life [3]. Psoriasis presents with erythematous, sharply demarcated scaly skin lesions that can cause pruritus, pain, and bleeding [4]. These plaques frequently appear on the surfaces of the knees, elbows, scalp, and back [2]. Psoriasis can pose a significant detriment to a patient’s quality of life (QoL), affecting their mental health, social life, and daily activities [1,3]. Although PsO is primarily a dermatological disease, its immune-mediated pathogenesis generates systemic effects and is closely linked with various comorbid conditions [2]. Psoriatic arthritis is the most common comorbidity, affecting close to 30% of psoriatic patients [2]. Other comorbidities associated with PsO include, but are not limited to, MetS, CVD, psychiatric disorders, inflammatory bowel disease, asthma, and non-alcoholic fatty liver disease [1,2,3,4,5]. The association between cardiometabolic comorbidities and PsO is particularly well established and involves certain cytokines and inflammatory molecules such as tumor necrosis factor (TNF)-alpha (α), interleukin (IL)-6, and IL-17 [6]. These cytokines are commonly associated with both PsO and MetS, suggesting the potential for targeted treatment in psoriatic patients with comorbidities. Although topical therapies remain the cornerstone for treating mild psoriasis, biologics that inhibit TNF-α, p40IL-12/23, IL-17, and p19IL-23, as well as oral phosphodiesterase 4 inhibitors, are considered first-line treatment options for moderate to severe plaque psoriasis when traditional drugs such as methotrexate and cyclosporin are contraindicated [7]. Apremilast is an oral inhibitor of phosphodiesterase 4 (PDE-4) that has been approved for the treatment of moderate-to-severe PsO and psoriatic arthritis [8]. These PDE-4 inhibitors increase levels of secondary messengers, such as intracellular cyclic adenosine monophosphate (cAMP), to downregulate pro-inflammatory cytokines and increase the expression of anti-inflammatory mediators such as IL-10 [9]. In addition to improvements in psoriatic disease activity, apremilast is reported to be associated with weight loss and improvements in hemoglobin A1c (HbA1c) serum levels; these effects are likely due to glucagon-like peptide-1 (GLP-1) activity [10,11,12,13]. Moreover, a preliminary study suggests potential vascular benefits of PDE-4 inhibition, with improved endothelial function in patients treated with apremilast [11,12]. Therefore, PDE-4 inhibition may have additional cardiometabolic effects beyond its anti-inflammatory role in psoriatic disease [13,14]. To evaluate this bivalent activity, we conducted a prospective observational study to evaluate the efficacy, safety, and tolerability of apremilast in patients with associated cardiometabolic comorbidities, characterizing dynamic changes in PsO clinimetric scores and metabolic profiles.

## 2. Results

### 2.1. Demographic Features of Enrolled Patients

In this observational study, 137 patients—57 females and 80 males—ranging in age from 36 to 89 years and affected by moderate-to-severe PsO, were enrolled from the Dermatology Unit at the Tor Vergata University Hospital. The study followed these patients for 52 weeks. The average age of enrolled patients was 63.7 (±13.3) years and the average body mass index (BMI) was 27.16 (±5.52) kg/m^2^, which is considered “pre-obese” [15]. A total of 30.7% of patients reported smoking at least 10 cigarettes a day, and 16.4% of patients reported daily alcohol use. After one year of treatment, 120 patients remained in the study as 17 subjects had dropped out: 5 subjects due to the ineffectiveness of the treatment, 9 subjects due to unfavorable side effects, 2 subjects lost at follow-up, and 1 subject due to a sudden rise in amylase levels (Table 1). The most commonly reported adverse effects (AEs) were gastrointestinal symptoms (nausea, diarrhea, vomiting) and general malaise, consistent with findings from Phase III studies [8,16].

### 2.2. Comorbidities of Enrolled Patients

At T0, each patient underwent a thorough physical examination and medical history interview during which comorbidities were reported. Of the 137 patients included at baseline, 22.1% presented with MetS (abdominal obesity, high blood pressure, impaired fasting glucose, high triglyceride levels, or low high-density lipoprotein (HDL) cholesterol levels) [17], 54.7% presented with hypertension (≥130/80 mmHg), 18.9% presented with DMII (fasting blood sugar levels ≥ 126 mg/dL), 45.3% presented with hypercholesterolemia (total cholesterol ≥ 200 mg/dL), 13.7% presented with hypertriglyceridemia (triglyceride levels ≥ 150 mg/dL) [18], and 29.5% presented with cardiovascular comorbidities. Moreover, 58% of our patients presented with psoriatic arthritis (PsA).

### 2.3. Assessing the Efficacy of Apremilast in the Context of Comorbid Conditions

After 52 weeks (T1), 120 patients were re-evaluated to investigate the safety and efficacy of apremilast in the context of comorbidities. An extensive medical history interview revealed no comorbidity-related exacerbations, thus confirming the high safety profile of this PDE-4 inhibitor. Furthermore, patients with comorbid conditions including hypercholesterolemia, hypertriglyceridemia, and MetS observed improvements in disease severity. In order to evaluate the effect of apremilast on hypercholesterolemia, a lipid profile was obtained from each patient at T0 and T1; all patients experienced a significant decrease in their lipid profiles after 12 months (Figure 1; *t*-test, *p* < 0.01). Moreover, patients on statin therapy observed a 15.78 point (7.56%) decrease in total cholesterol, suggesting a synergistic activity with apremilast treatment. Patients who were not on statin therapy observed a 6.26 point (3.44%) decrease in total cholesterol. The improvement in total cholesterol among patients not on statins supports the efficacy of apremilast in improving total cholesterol. These results highlight how apremilast can modulate total serum cholesterol levels in patients independently of comorbidities, and how prolonged treatment can influence these parameters. Moreover, of the 43 subjects affected by hypercholesterolemia at T0, 15 (34.8%) returned at T1 with a low-density lipoprotein (LDL) cholesterol below 129 mg/dL, which is within the normal range of values [19]. The cholesterol profiles of the 40 subjects on statin therapy at T0 and T1 were also compared to account for the role of statins in treating hypercholesterolemia. The cholesterol profiles of the 40 subjects on statin therapy at T0 and T1 were also compared to account for the role of statins in treating hypercholesterolemia. The results showed that while there were no significant influences on high-density lipoprotein (HDL) levels, LDL values decreased in the total population (repeated measures ANOVA test, *p* < 0.01), particularly among statin-treated patients (repeated measures ANOVA test, *p* < 0.001), highlighting the synergistic effects of apremilast in combination with statins in psoriatic patients (see Figure 2 and Figure 3).

The lipid profile obtained from patients at T0 and T1 was also evaluated to assess the effect of apremilast on hypertriglyceridemia. Of the 13 subjects affected by hypertriglyceridemia at T0, 8 (61.5%) returned at T1 with triglyceride levels below 150 mg/dL (1.7 mmol/L), which represents a normal triglyceride value [20]. Moreover, when considering the average triglyceride values in the study population, reductions were statistically significant in both the total population (repeated measures ANOVA test, *p* < 0.01) and in patients treated with statins (Figure 4; repeated measures ANOVA test, *p* < 0.001).

The effect of apremilast on glycemia was compared from T0 to T1 by analyzing the glucose mean serum level. Among the 18 diabetic patients receiving insulin and/or hypoglycemic therapy for the duration of the study, a 10.8 point (8.2%) decrease in glycemic values from T0 to T1 was observed. At T0, the average glycemic value was 132 mg/dL, while at T1 the average glycemic value was 121.17 mg/dL, suggesting a synergistic action of apremilast among patients on insulin and/or hypoglycemic therapy (Figure 5; repeated measures ANOVA test, *p* < 0.05). Among the 102 patients not receiving insulin and/or hypoglycemic therapy, no significant changes in glycemic values were observed; at T0 the glycemic value was 87.95 mg/dL, while at T1 the glycemic value was 88.31 mg/dL.

Despite a reduction in average blood glucose levels, there were no differences observed in the average HbA1c values across the study population, irrespective of insulin treatment (Figure 6).

Considering all the biochemical parameters evaluated, twenty-one patients were affected by MetS at T0. After 52 weeks, six patients (28.5%) achieved control of both hypercholesterolemia and hypertriglyceridemia, no longer falling into the aforementioned category (Figure 7; McNemar’s test; *p* < 0.05).

### 2.4. Assessing the Efficacy of Apremilast in Psoriasis Management

After 52 weeks, the 120 remaining patients were re-evaluated to investigate the role of apremilast in psoriatic disease. The Psoriasis Area and Severity Index (PASI), Dermatology Life Quality Index (DLQI), Nail Area Psoriasis Severity Index Score (NAPSI), Itching Intensity on a 10 cm visual analog scale (VAS), and Tender Joint Count (TJC) scores were measured at T0 and T1; all criteria demonstrated a statistically significant decrease without the influence of gender and age (Figure 4; *t*-test; *p* < 0.001). At T0, an average PASI score of 10.6 was observed. After 52 weeks of apremilast treatment, there was a 7.9 point (75.2%) decrease in PASI score, setting the average T1 PASI score at 2.6. According to the Food and Drug Administration (FDA), effective psoriasis treatment is defined as a 75% reduction in the PASI score; apremilast treatment in the context of this study yielded a 75.2% reduction in the average PASI score, deeming it a successful treatment method (Figure 8 and Figure 9). At T0, an average NAPSI score of 52 was observed. After 52 weeks of apremilast treatment, there was a 3.62 point (27%) decrease in the average NAPSI score, setting the average T1 NAPSI score at 5 (Figure 8). At T0, the average Pain VAS score was 45.2, and by the end of the 52-week treatment period, the Pain VAS score decreased by 26.6 points (58.8%). At T0, the average TJC score was 8.28; at T1 the average TJC score was 3.81, marking a 4.47 point (54%) decrease from baseline. Finally, at T0, an initial DLQI value of 8.62 was observed; by week 52 of apremilast treatment, there was a 3.62 point (42%) decrease from baseline (Figure 8). In summary, the reduction in the PASI, NAPSI, DLQI, and PAIN VAS indexes was statistically significant (one-way ANOVA repeated measures; *p* < 0.001). No significant statistical variations were found between comorbidities and clinimetric scores.

The serum erythrocyte sedimentation rate (ESR) and C-reactive protein (CRP) are two laboratory diagnostic tests that help provide valuable information regarding inflammation and infection [21]. Both ESR and CRP demonstrated a statistically significant decrease by the end of the 52-week evaluation period (Figure 10). The ESR was 28.24 mm/hour at T0 and decreased to 21 mm/hour at T1 (one-way ANOVA repeated measures; *p* < 0.05). Likewise, the CRP was 4.76 mg/dL at T0 and decreased to 3.22 mg/dL at T1 (repeated measures ANOVA test; *p* < 0.05).

### 2.5. Safety and Tolerability

At the onset of the study, 137 enrolled patients began apremilast treatment at the Department of Dermatology at the Tor Vergata University Hospital. By the end of the 52-week trial, 17 patients were removed from the study (dropout). Of these patients, nine terminated apremilast treatment due to possible drug intake reactions. Specifically, eight patients reported gastrointestinal symptoms including nausea, vomiting, and diarrhea, while one patient reported a rise in serum amylase levels. These AEs were not statistically significant (chi-square test; *p* > 0.05) in the study population, confirming the safety profile of apremilast treatment and aligning with findings from previous trials [8,16].

## 3. Discussion

Psoriasis is a multifactorial condition often associated with comorbidities such as CVD, MetS, and DMII [22]. Prior studies indicate a correlation between PsO and increased risk of MetS.

Inflammatory mediators have the potential to impact a wide range of diseases, including obesity and its associated MetS. Obesity has been identified as a risk factor for psoriasis, and likewise, a trend in weight increase in patients with PsO has been observed [23,24]. This bidirectional risk is speculated to be caused by the overlap of inflammatory cytokines involved in both diseases, cellular sources of inflammatory cytokines, and alterations in oxidative stress levels [25]. Among the different factors that play a role in inflammation, TNF-α, IL-6, IL-17, and IL-10 are cytokines that orchestrate both PsO and MetS. The IL-17 family participates in the complex interplay between inflammation and metabolism, with systemic effects on glucose homeostasis and a negative regulatory role in adipogenesis and adipocyte function [26]. Moreover, obesity has been shown to promote the expansion of IL-17-producing T cells in adipose tissue, inducing a vicious cycle in which IL-17 promotes inflammation through a positive feedback mechanism. Moreover, treatment of human keratinocytes with palmitic acid, a fatty acid mainly involved in obesity, induces the expression of Th17 cell-related cytokines with Regenerating islet-derived protein 3 gamma (Reg3γ), which results in epidermal hyperplasia, like psoriasis [27].

Moreover, MetS results from the altered response of immunity and macrophage infiltration of adipose tissue. TNF-α and IL-6 are pro-inflammatory cytokines secreted from the peri-visceral fat; their effect is linked to insulin resistance, atherosclerosis, and endothelial dysfunction [28,29]. TNF-α promotes carbohydrate dysregulation (hyperglycemia) by inhibiting insulin action, reducing glucose clearance (primarily by muscle and adipose tissue), and increasing hepatic glucose production. The promotion of hyperlipidemia by TNF-α is primarily mediated by stimulation of hepatic lipid synthesis [30] and adipose lipolysis, along with suppression of triacylglycerol clearance and inhibition of insulin-stimulated de novo lipogenesis (in adipose tissue) [31]. Psoriasis is characterized by a high serum level of TNF-α, linking PsO and MetS. In addition to TNF-α, IL-6 acts on adipose tissue to increase leptin secretion, suppress satiety, and increase adipose tissue lipolysis, which, in turn, promotes hepatic gluconeogenesis and hepatic insulin resistance [32,33]. Considering apremilast’s role in reducing IL-17F, IL-17A, IL-22, and TNF-α plasma levels in patients with moderate to severe plaque psoriasis, it could be hypothesized that apremilast acts as a pleiotropic molecule, blocking a common pathway shared by both PsO and MetS. Conversely, low IL-10 levels are associated with MetS [34]. IL-10 is a paramount anti-inflammatory cytokine that is directly involved in fat metabolism. Specifically, the fatty acid desaturation induced by IL-10 reconfigures the abnormal activation of NF-κβ family transcription factors (such as REL) and the accumulation of saturated very long chain (VLC) ceramides. These findings support the idea that the regulation of fatty acid homeostasis in innate immune cells serves as a critical regulatory mechanism for controlling pathological inflammation. This also implies that “metabolic correction” of VLC homeostasis may be a crucial tactic in restoring dysregulated inflammation from IL-10 deficiency [34,35]. Apremilast’s ability to increase intracellular IL-10 levels plays an important role in MetS and paves the way for future studies on the association between PsO and MetS.

Apremilast is an oral PDE-4 inhibitor that has shown efficacy in the treatment of PsO, as it modulates intracellular signaling pathways, primarily through the reduction in the aforementioned pro-inflammatory cytokines [21,22]. Various studies have explored the potential benefits of apremilast in treating MetS [36,37,38]. It is hypothesized that apremilast may have a positive impact on factors associated with MetS, such as insulin resistance and lipid metabolism [36], as PDE-4-mediated cAMP signaling has a critical role in the context of glucose and lipid metabolism [36,37]. Recent studies also suggest that apremilast may act on the sirtuin pathway, possessing a potential anti-atherosclerotic effect [39]. Sirtuin 1 (SIRT1) is a protein encoded by the *SIRT1* gene that participates in the efflux of cholesterol. It has been demonstrated that apremilast increases SIRT1 levels, decreasing total cholesterol and the risk of developing atherosclerotic plaques [39,40]. Furthermore, apremilast inhibits PDE-4, which increases cAMP levels and activates protein kinase A (PKA); this activity is responsible for the phosphorylation of downstream effectors such as cAMP-responsive element binding protein (CREB) and inactivation of nuclear factor kappa B (NF-κB) [41,42]. The elevated cAMP also activates hormone-sensitive lipase (HSL), which enhances lipolysis [43]. In the pooled analysis of phase 3 ESTEEM and PALACE and phase 3b LIBERATE trials, patients on antidiabetic medications and apremilast treatment demonstrated greater reductions in HbA1c than patients receiving a placebo [43,44]. Patients on apremilast with baseline HbA1c ≥ 5.7% demonstrated greater reductions in HbA1c than placebo patients [45]. Moreover, Mazzilli et al. [46] identified an improved response in PsO clearance in patients with concomitant diabetes, underlying the bidirectional interplay between these disease states. In our 12-month observational study, we observed positive outcomes not only in psoriatic disease but also in markers of MetS. Baseline subjects had a mean age of 63.7 years, weight of 79 kg, height of 169 cm, and BMI of 27.45 kg/m^2^. Among the participants, 22.1% had MetS, 54.7% had arterial hypertension, 18.94% had DMII, 45.26% had hypercholesterolemia, and 29.47% had cardiovascular comorbidities. At the time of recruitment, most of the patients were following one or more concomitant pharmacological treatments. Specifically, 18.94% received treatment with insulin and/or other hypoglycemic drugs, 42.1% with statins, 30.5% with anticoagulants, 10.5% with thyroid hormones, and 49.47% with antihypertensives. One of the most intriguing findings among the various comorbid conditions studied was the significant role of apremilast in modulating the lipid profile. Twelve months of apremilast treatment provided a significant reduction in total and LDL cholesterol levels among patients with PsO and concomitant hypercholesterolemia. Changes in total cholesterol values among statin and non-statin users were also correlated; after 52 weeks of apremilast treatment, the total and LDL cholesterol decreased by 7.56% and 3.44%, respectively. Moreover, although no significant alterations were reported in average HDL values, apremilast has a demonstrated impact on LDL levels in the study population. Specifically, apremilast has been shown to improve oxidized LDL-induced endothelial dysfunction by restoring Krüppel-like-factor-6 expression and has beneficial metabolic effects, suggesting a potential role for apremilast in improving cardiovascular function [47,48].

Furthermore, diabetic patients on insulin and/or hypoglycemic therapy observed a decrease in glycemic values from 132 mg/dL at T0 to 121.17 mg/dL at T1. These results confirm previous findings and contribute to the growing understanding of the role of apremilast in PsO management and its potential in managing MetS [10]. In line with the reduction in cholesterol, average triglyceride values also decreased in the total population, especially in the statin-treated patients. This paves the way to consider apremilast as a pleiotropic molecule that can synergize with statins in the control of dysmetabolism, especially in psoriatic patients with MetS, and confirms previous studies where a reduction in triglycerides has been reported [49]. The secondary objective of this study was to evaluate the efficacy and safety of apremilast among patients with PsO. Clinimetric indices such as PASI, Pain VAS, TJC, DLQI, and NAPSI were used to measure disease progression. At baseline, an average PASI score of 10.5 was observed; by week 52, the average PASI score decreased to 2.6, as represented in Figure 8. The Pain VAS score and TJC also demonstrated a reduction from baseline values; the Pain VAS score decreased from 45.2 at T0 to 18.6 at T1, and the TJC decreased from 8.28 at T0 to 3.81 at T1. The DLQI demonstrated a slight reduction from 8.62 at T0 to 5 at T1, suggesting a mild improvement in the average patient QoL. Finally, the NAPSI score demonstrated the high efficacy of apremilast treatment, with the average score decreasing from 5 at T0 to 3.62 by the end of the 52-week trial period. Our results are consistent with previously published findings, demonstrating apremilast’s role as an effective treatment for moderate PsO, especially in the nail form [46,47,48,49,50,51,52]. Furthermore, the study illustrates apremilast’s efficacy in improving QoL, thereby reducing the overall disease burden.

## 4. Materials and Methods

All patients were enrolled from the Dermatology Unit at the Tor Vergata University Hospital in Rome, Italy. The study included patients 18 years and older with moderate-to-severe PsO (defined as a PASI score of ≥10) who were either refractory to topical therapies, or had contraindications to traditional systemic therapies and biologics. These patients presented with concomitant comorbidities, such as CVD or MetS. Each patient underwent appropriate wash-out from previous PsO therapies and signed an informed consent form prior to the initiation of apremilast therapy. Certain exclusion criteria were implemented to ensure patient safety and the study’s validity. Patients with anorexia, congenital or acquired immunodeficiencies, or galactose intolerance were excluded from the study due to potential adverse reactions. Pregnant or breastfeeding patients were also excluded from the study to protect the mother and fetus from harm. Prior to the administration of apremilast, each subject underwent a thorough physical examination and medical history interview. At baseline (T0), each patient was evaluated using the following clinical measurements: PASI, DLQI, NAPSI, Itching Intensity on a 10 cm VAS, TJC. After 52 weeks (T1), the clinical measurements for each patient were re-evaluated to investigate the efficacy and safety of apremilast in patients with comorbidities. Moreover, blood analysis, cholesterol, triglycerides, glucose, HbA1c, and CRP values were evaluated at each visit. Ethical review and approval for this study were in accordance with relevant national legislation, specifically legislative decree 81/2008.

### Statistical Analysis

All statistical data were initially entered into Microsoft Excel (Version 2406, Microsoft, Redmond, WA, USA) and analysis was performed using Windows Social Sciences Statistics Package, version 20.0 (SPSS, Chicago, IL, USA). A comparison of normal variables between pre- and post-treatment groups was performed with a paired student’s *t*-test. The chi-squared (chi^2^) test was used to compare dichotomous data reported as percentages. All respective *p* values < 0.05 were considered statistically significant and all graphs were produced using Microsoft Excel.

## 5. Conclusions

When selecting the optimal therapeutic approach for treating PsO, clinicians must consider the severity of the clinical presentation, the repercussions on the QoL, and the comorbidities that may be affected by the treatment option [53]. The emergence of PDE-4 inhibitors such as apremilast provides an effective, safe, and tolerable treatment for psoriatic patients, especially those with associated metabolic comorbidities. After 52 weeks of apremilast treatment, patients experiencing comorbid conditions, including hypercholesterolemia, hypertriglyceridemia, hyperglycemia, and MetS, achieved improvements in comorbid disease severity. Our results even suggest a possible synergistic activity among apremilast and statin and/or insulin/hypoglycemic therapy. Furthermore, the PASI, Pain VAS, TJC, DLQI, NAPSI, ESR, and CRP scores measured at T0 and T1 demonstrated a high efficacy and safety profile of apremilast through a statistically significant decrease in psoriatic disease severity. While the results of our study require validation on a larger scale, the use of apremilast in treating psoriatic patients with cardio-metabolic comorbidities yields promising potential for addressing both conditions simultaneously.

## Figures and Tables

**Figure 1 pharmaceuticals-17-00989-f001:**
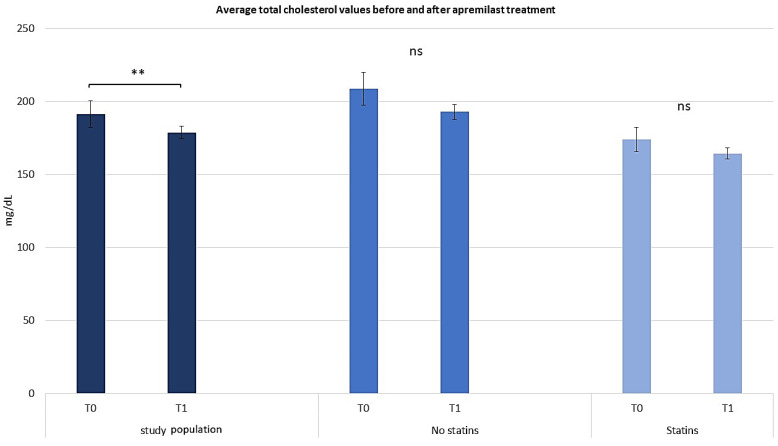
Graphical representation of average total cholesterol values before and after apremilast treatment. The graphs show the mean ± SD values. ** *p* < 0.01. ns, no significance (repeated measures ANOVA test).

**Figure 2 pharmaceuticals-17-00989-f002:**
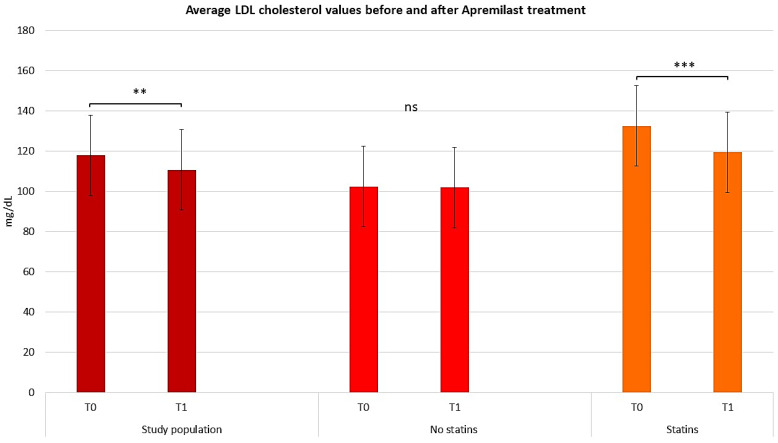
Graphical representation of average LDL cholesterol values before and after apremilast treatment for the study population, patients treated with and without statins. The graphs show the mean ± SD values. ** *p* < 0.01, *** *p* < 0.001. ns, no significance (repeated measures ANOVA test).

**Figure 3 pharmaceuticals-17-00989-f003:**
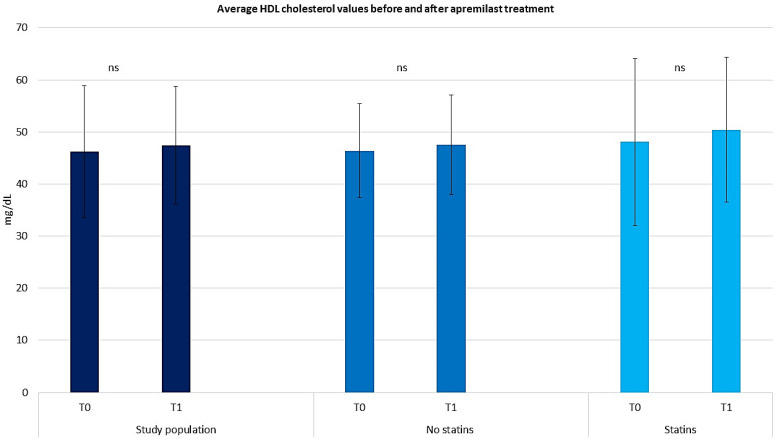
Graphical representation of average HDL cholesterol values before and after apremilast treatment for the study population, patients treated with and without statins. The graphs show the mean ± SD values. ns, no significance (repeated measures ANOVA test).

**Figure 4 pharmaceuticals-17-00989-f004:**
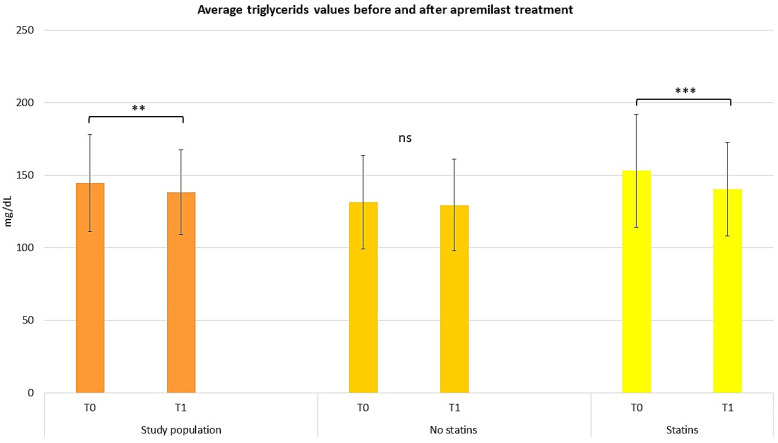
Graphical representation of average triglycerides values before and after apremilast treatment for the study population, patients treated with and without statins. The graphs show the mean ± SD values. ** *p* < 0.01, *** *p* < 0.001. ns, no significance (repeated measures ANOVA test).

**Figure 5 pharmaceuticals-17-00989-f005:**
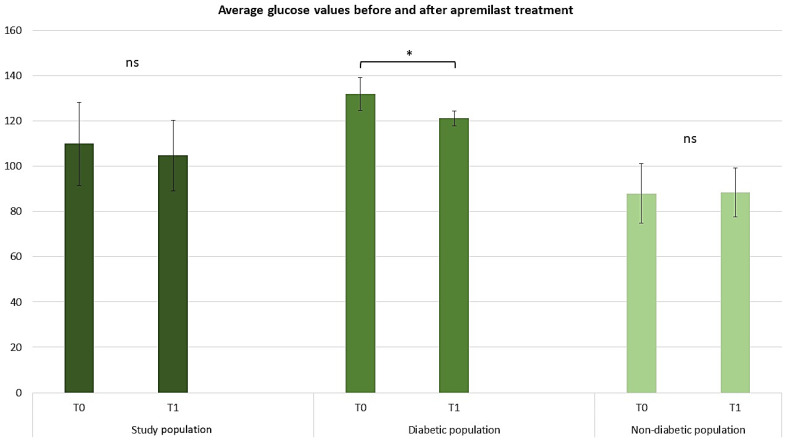
Graphical representation of average blood glucose values before and after treatment in diabetic versus non-diabetic patients. The graphs show the mean ± SD values. * *p* < 0.05. ns, no significance (repeated measures ANOVA test).

**Figure 6 pharmaceuticals-17-00989-f006:**
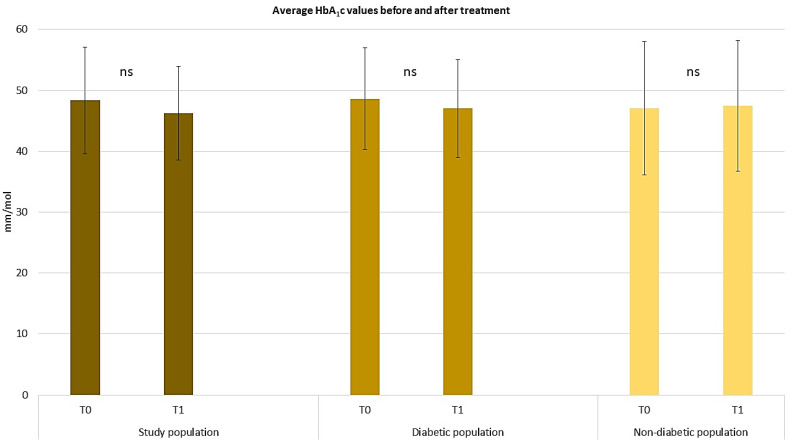
Graphical representation of average HbA_1_c values before and after treatment in diabetic versus non-diabetic patients. The graphs show the mean ± SD values. ns, no significance (repeated measures ANOVA test).

**Figure 7 pharmaceuticals-17-00989-f007:**
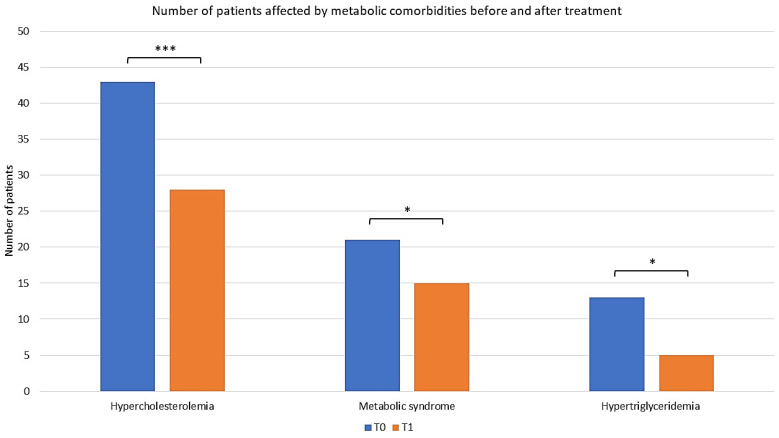
Distribution of patients with metabolic syndrome, hypercholesterolemia, and hypertriglyceridemia before and after apremilast treatment. The graphs show the number of patients before and after treatment. * *p* < 0.05, *** *p* < 0.001 (McNemar’s test).

**Figure 8 pharmaceuticals-17-00989-f008:**
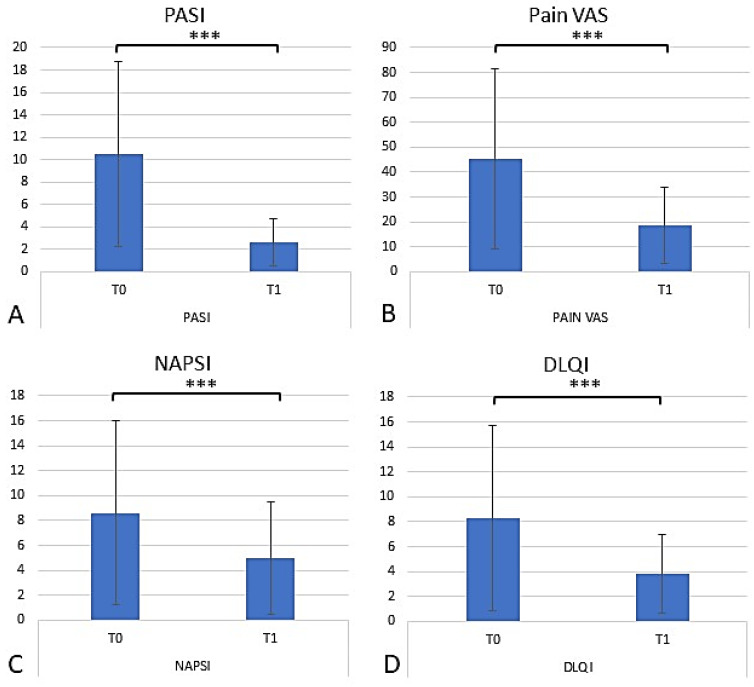
Variation in T0 and T1 clinimetric indices among patients on apremilast therapy. (**A**–**D**) Reduction in the PASI, NAPSI, DLQI, and PAIN VAS indexes was statistically significant. The graphs show the mean ± SD values. *** *p* < 0.001 (repeated measures ANOVA test).

**Figure 9 pharmaceuticals-17-00989-f009:**
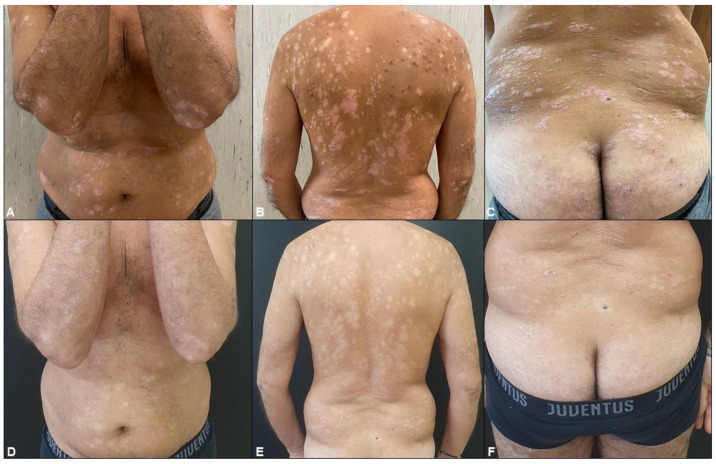
Clinical evaluation at T0 and after 36 weeks of treatment: (**A**–**C**) Clinical examination revealed severe plaque psoriasis in the context of metabolic syndrome. (**D**–**F**) Clinical improvement of psoriasis and BMI after 36 weeks of treatment with apremilast.

**Figure 10 pharmaceuticals-17-00989-f010:**
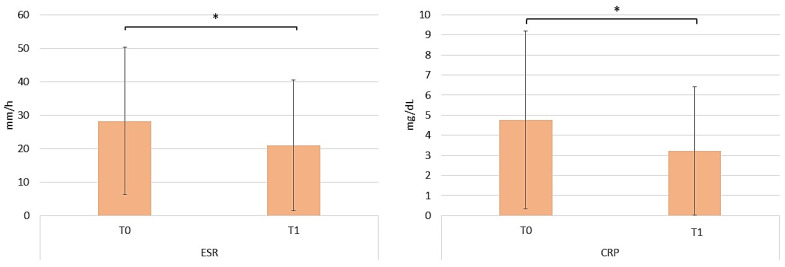
Variation in ESR and CRP in patients on apremilast therapy from baseline to T1. The graphs show the mean ± SD values. * *p* < 0.05 (repeated measures ANOVA test).

**Table 1 pharmaceuticals-17-00989-t001:** Summary of Patient Demographics, Comorbidities, and Pharmacological Treatments.

Parameter	Number	Percentage of Patients
Demographics		
Male	80	58.4%
Female	57	41.6%
Average Age (range), years	63.7 (22–87)	
Average Height (range), cm	169 (150–206)	
Average Weight (range), kg	78.4 (48–155)	
Average BMI (range) $,\;\mathrm{kg}{/m}^{2}$	27.16 (16.7–50.6)	
Number of Patients at T0	137	100%
Number of Patients at T1	120	87.59%
Patient Dropout	17	12.40%
Previous Dermatological Treatments	122	89.05%
Topical treatments	40	32.79%
Cyclosporine	36	29.51%
Methotrexate	24	19.67%
Acitretin	6	4.92%
Narrowband-UVB	6	4.92%
Dimethyl fumarate	3	2.46%
Adalimumab	4	3.28%
Etanercept	3	2.46%
Comorbidities		
Metabolic Syndrome	21	22.1%
Hypertension	52	54.7%
Diabetes Mellitus	18	18.9%
Hypercholesterolemia	43	45.3%
Hypertriglyceridemia	13	13.7%
Cardiovascular Comorbidities	28	29.5%
Psoriatic Arthritis	80	58%
Pharmacological Treatments		
Insulin and/or Hypoglycemics	18	18.9%
Statins	40	42.1%
Anticoagulants	29	30.5%
Antihypertensives	47	49.5%

## Data Availability

All data generated or analyzed during this study are included in this article. Further inquiries can be directed to the corresponding author.

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
