# Peer review of "Apremilast as a Potential Targeted Therapy for Metabolic Syndrome in Patients with Psoriasis: An Observational Analysis"

_pharmaceuticals, 2024, doi:10.3390/ph17080989_

Round 1

Reviewer 1 Report

Comments and Suggestions for Authors

This paper is regarding the possible effects of apremilast against metabolic syndrome in patients with psoriasis. 
The paper is mostly well written. However, additional results and discussion are required for publication:

The effects of premolar on LDL-cholesterol, HbA1c, systolic or diastolic blood pressure should be examined and shown.

In discussion, the effects of apremilast to alter the production of TNFa, IL-6, IL-17 or IL-10 might be related to the anti-metabolic syndrome effects. This discussion should be added.

Comments on the Quality of English Language

No specific comments.

Author Response

First of all,

we would like to thank the reviewer for his/her constructive comments.

Below are our answers, item by item.

With our best regards

Elena Campione and Co-authors

Reviewer 1

1.This paper is regarding the possible effects of apremilast against metabolic syndrome in patients with psoriasis. The paper is mostly well written. However, additional results and discussion are required for publication. Dear Reviewer, thank you for your interest in our manuscript and for the profitable comments to improve its quality. 1. The effects of premolar on LDL-cholesterol, HbA1c, systolic or diastolic blood pressure should be examined and shown.

1.Dear reviewer, thank you for the valuable comment. We have added graphics for triglyceride, LDL, and HbA1c values to clarify this point. Systolic and/or diastolic blood pressure values where not examined during the study period and hypertension was only investigated in the anamnesis, at baseline and at week 52.

2.In discussion, the effects of apremilast to alter the production of TNFa, IL-6, IL-17 or IL-10 might be related to the anti-metabolic syndrome effects. This discussion should be added.

2.Dear reviewer, thank you for the valuable comment. We have reported and improved this point in the discussion section as follows:

“Inflammatory mediators have the potential to impact a wide range of diseases, including obesity and its associated MetS. Obesity has been identified as a risk factor for psoriasis, and likewise, a trend in weight increase in patients with psoriasis has been observed [21,22]. This bidirectional risk is speculated to be caused by the overlap in inflammatory cytokines involved in both diseases, cellular sources of inflammatory cytokines, and alterations in oxidative stress levels [23]. Among the different factors that play a role in inflammatory, TNF-α, IL-6, IL-17, and IL-10 are cytokines that orchestrate both PsO and MetS. The IL-17 family participates in the complex interplay between inflammation and metabolism, with systemic effects on glucose homeostasis and a negative regulatory role in adipogenesis and adipocyte function [24]. Moreover, obesity has been shown to promote the expansion of IL-17-producing T cells in adipose tissue, inducing a vicious cycle in which IL-17 promotes inflammation through a positive feedback mechanism. Moreover, treatment of human keratinocytes with palmitic acid, a fatty acid mainly involved in obesity, induces the expression of Th17 cell-related cytokines with Regenerating islet-derived protein 3 gamma (Reg3γ), which results in epidermal hyperplasia, like psoriasis [25]. Moreover, MetS results from the altered response of immunity and macrophage infiltration of adipose tissue. TNF-α and IL-6 are pro-inflammatory cytokines secreted from the peri-visceral fat; their effect is linked to insulin resistance, atherosclerosis, and endothelial dysfunction [26,27]. TNF-α promotes carbohydrate dysregulation (hyperglycaemia) by inhibiting insulin action, reducing glucose clearance (primarily by muscle and adipose tissue) and increasing hepatic glucose production. Promotion of hyperlipidaemia by TNF-α is primarily mediated by stimulation of hepatic lipid synthesis [28] and adipose lipolysis, along with suppression of triacylglycerol clearance and inhibition of insulin-stimulated de novo lipogenesis (in adipose tissue) [29]. Psoriasis is characterized by high serum levels of TNF-α, linking PsO and MetS. In addition to TNF-α, IL-6 acts on adipose tissue to increase leptin secretion, suppress satiety, and increase adipose tissue lipolysis, which, in turn, promotes hepatic gluconeogenesis and hepatic insulin resistance [30,31]. Considering apremilast’s role in reducing IL-17F, IL-17A, IL-22, and TNF-α plasmatic levels in patients with moderate to severe plaque psoriasis, it could be postulated that apremilast acts as a pleiotropic molecule, blocking the common pathway shared by PsO and MetS. Conversely, low IL-10 levels are associated with MetS [32]. IL-10 is a paramount anti-inflammatory cytokine that is directly involved in fat metabolism. Specifically, the fatty acid desaturation programme induced by IL-10 rewires the abnormal activation of NF-κβ family transcription factors (REL) and the build-up of saturated very long chain (VLC) ceramides. These findings support the notion that innate immune cells' fatty acid homeostasis functions as a crucial regulatory node to regulate pathologic inflammation. This also implies that "metabolic correction" of VLC homeostasis may be a crucial tactic in restoring dysregulated inflammation from IL-10 deficiency [32,33]. Apremilast’s ability to increase intracellular IL-10 levels play an important role in MetS and pave the way for future studies on the association between PsO and MetS.”

We appreciate all your feedback and have carefully considered your suggestions to improve our manuscript.

Reviewer 2 Report

Comments and Suggestions for Authors

The author conducted a prospective observational study to evaluate the efficacy and safety of apremilast for metabolic and cardiovascular comorbidities in psoriasis patients. They found that apremilast improved metabolic and cardiovascular comorbidities with durable safety profiles along with the relief of psoriasis symptoms. I have some concerns.

1. Please add the information on prior therapies in Table 1. Was apremilast the first systemic therapy in all patients. Prior systemic therapies might affect the results of this study.

2. No significant differences in total cholesterol values before and after apremilast treatment in statin group and non-statin group in Figure 1? In addition, based on the present data, the decrease in total cholesterol values seemed to be more significant in non-statin group than statin group. Are labels correct?

3. No statistical bars exist in Figure 2.

4. The labels of X-axis in Figure 3, “Insulin” and “No insulin” should be replaced by “diabetic population” and “non-diabetic population”.

5. The font size of labels of X-axis and Y-axis in many figures is too small. Figures are not designed to be understandable to the reader and must be replaced by the sophisticated ones.

6. The authors mentioned, “Both ESR and CRP significantly decreased by the end of the 52-week evaluation (Figure 6).”. However, results of statistical analysis were shown neither in Figure nor manuscript.

7. The details of unfavorable side effects in 9 patients who suspended apremilast therapy should be provided.

8. I am interested in whether the decrease in total cholesterol values or blood glucose values was correlated with the decrease in psoriasis severity scores.

9. The number and age of enrolled participants should be presented only in Results section but not in Materials and Methods section.

10. Some abbreviations were used without the explanation. Moreover, abbreviations should be explained when it is used first in the manuscript but not in the Materials and Methods section.

11. Some unnecessary capitalizations, such as “Tumor necrosis factor” and “Apremilast”, are observed. Please check the manuscript carefully.

Comments on the Quality of English Language

Please see above.

Author Response

First of all,

we would like to thank the reviewer for his/her constructive comments. Below are our answers, item by item.

With our best regards

Elena Campione and Co-authors

Reviewer 2

The author conducted a prospective observational study to evaluate the efficacy and safety of apremilast for metabolic and cardiovascular comorbidities in psoriasis patients. They found that apremilast improved metabolic and cardiovascular comorbidities with durable safety profiles along with the relief of psoriasis symptoms. I have some concerns.

Dear Reviewer, thank you for your interest in our manuscript and for your valuable comments aimed at improving its quality.

  1. Please add the information on prior therapies in Table 1. Was apremilast the first systemic therapy in all patients. Prior systemic therapies might affect the results of this study.

Dear reviewer, we appreciate your valuable feedback. In the Materials and Methods section, we specify that patients included in the study were either refractory to topical therapies or had contraindications or failure to traditional systemic therapies and biologics. However, we ensured appropriate wash-out from previous psoriasis therapies, as discussed in the Materials and Methods section. We have now included this clarification in Table 1 to address this feedback.  

  1. No significant differences in total cholesterol values before and after apremilast treatment in statin group and non-statin group in Figure 1? In addition, based on the present data, the decrease in total cholesterol values seemed to be more significant in non-statin group than statin group. Are labels correct?

Dear reviewer, thank you for the valuable comment. We have improved the figure quality and updated the statistical analyses included total cholesterol, LDL and HDL. No statistically differences (p< 0.05) were found in statins vs non-statins patients but just after treatment in the study population.

  1. No statistical bars exist in Figure 2.

Dear reviewer, thank you for the valuable comment. Figure 2 reports the absolute number of patients affected by comorbidities before and after treatment, not mean or median values, so statistical bars for SD or SEM cannot be added to the histograms. However, we reported the statistical significance of improving this point.

  1. The labels of X-axis in Figure 3, “Insulin” and “No insulin” should be replaced by “diabetic population” and “non-diabetic population”.

Dear reviewer, thank you for the valuable comment. We agree with your feedback and have changed the labels in Figure 3 to reflect this suggestion. 

  1. The font size of labels of X-axis and Y-axis in many figures is too small. Figures are not designed to be understandable to the reader and must be replaced by the sophisticated ones.

Dear reviewer, thank you for your feedback. We have increased the font sizes of the X and Y- axes in our graphs as per your suggestion. These changes aim to enhance readability and understanding for the reader.

  1. The authors mentioned, “Both ESR and CRP significantly decreased by the end of the 52-week evaluation (Figure 6).”. However, results of statistical analysis were shown neither in Figure nor manuscript.

Dear reviewer, we appreciate the feedback. We agree with your assessment and have added the missing information required.

  1. The details of unfavorable side effects in 9 patients who suspended apremilast therapy should be provided.

Dear reviewer, thank you for the comment. As discussed in section 2.1 “Demographic Features of Enrolled Patients,” the most common adverse effects reported were gastrointestinal symptoms and general malaise, in line with the Phase III trial. We have added another line to this section to clarify this point. Thank you for the suggestion.

  1. I am interested in whether the decrease in total cholesterol values or blood glucose values was correlated with the decrease in psoriasis severity scores.

Dear reviewer, thank you for the valuable comment. Multiparametric analyses have not shown correlations between blood values and clinimetric scores.

  1. The number and age of enrolled participants should be presented only in Results section but not in Materials and Methods section.

Dear reviewer, thank you for the valuable comment. We have removed the number and age of enrolled participants from the Materials and Methods section and added it to the Results section.

  1. Some abbreviations were used without the explanation. Moreover, abbreviations should be explained when it is used first in the manuscript but not in the Materials and Methods section.

Dear reviewer, thank you for the valuable comment. We have edited the manuscript to reflect this point.

  1. Some unnecessary capitalizations, such as “Tumor necrosis factor” and “Apremilast”, are observed. Please check the manuscript carefully.

Dear reviewer, thank you for the feedback. We have reviewed the manuscript and made appropriate changes to address unnecessary capitalizations.

We appreciate all of your feedback and have carefully considered your suggestions to improve our manuscript.

Reviewer 3 Report

Comments and Suggestions for Authors

In this observational study by Bianchi and colleagues, the authors evaluate the effect of oral apremilast administration on psoriasis patients followed up over a period of one year. Apremilast is a small molecule inhibitor of phosphodiesterase 4 (PDE4), which increases intracellular cAMP concentration leading to the suppression of inflammation. Herein, in addition to evaluating the efficacy of Apremilast on psoriasis disease, authors have also investigated its effect on other co-morbidities associated with psoriasis. All the 120 psoriasis patients followed up, had other co-morbidities such as hypertension, hypercholesterolemia, hypertrigeleridemia, Diabetes, CVD and psoriatic arthritis. Key results that has emerged out of this study include, a) A significant decrease in patients serum cholesterol, b) Substantial decrease in no of patients with metabolic syndrome, hypercholesterolemia and hypertriglyceridemia, c) Significant decrease in average blood glucose concentration in those patients who is also taking insulin, d) Signiant decrease in PASI, Pain, Nail pittance and DLQI values. The anti-inflammatory action of apremilast administration was supported by a significant decrease in serum inflammatory marker, CRP erythrocyte sedimentation rate. Overall these results indicate that Apremilast can either have a synergistic effect on the ongoing treatment regimen for the co-morbidities.  These results highlight, not only the clinical efficacy of apremilast on psoriasis but also on the other co-morbidities warranting publication in this journal but some minor queries are placed here.

Comments:

1.     Only serum CRP data is provided for the anti-inflammatory action of apremilast. Could the authors provide supporting data on serum TNFa/IL17 given that TNFa is also a global inflammatory marker involved in other comorbidities while IL17 is directly relavent to psoriasis pathogenesis?

2.     In Figure 1, even without statin co-treatment there is an apparent reduction in total cholesterol levels. Did the authors perform statistical analysis on this data (since I do not see any n.s. on top of this data)? If the decrease observed is significant, it further strengthen the innate anti-inflammatory activity of apremilast influencing total cholesterol levels.

3.     Why authors did not include the total triglyceride, in patients before and after treatment of apremilast, similar to Figure 1 but instead, they provide the no of patients who have recovered from these co-morbidities. As a reviewer/reader, I would be interested to see the extent of triglyceride reduction after apremilast treatment with and without statins.

4.     Pairwise statistical analysis need to be included for Figure 6.

5.     Figure 5 legend: I think its T0 not W0

6.     Abbreviations such as PASI, NAPSI, VAS, TJC, DLQI need to be expanded at their first use in results. Though this has been expanded in methods, this section comes after the results and discussion

7.     I do not see the supplementary files online. Have they been uploaded?

8.     No details on the ethical approvals obtained for this study is mentioned  (is it in the supplementary files?)

Author Response

First of all,

we would like to thank the reviewer for his/her constructive comments. Below are our answers, item by item.

With our best regards

Elena Campione and Co-authors

Reviewer 3

In this observational study by Bianchi and colleagues, the authors evaluate the effect of oral apremilast administration on psoriasis patients followed up over a period of one year. Apremilast is a small molecule inhibitor of phosphodiesterase 4 (PDE4), which increases intracellular cAMP concentration leading to the suppression of inflammation. Herein, in addition to evaluating the efficacy of Apremilast on psoriasis disease, authors have also investigated its effect on other co-morbidities associated with psoriasis. All the 120 psoriasis patients followed up, had other co-morbidities such as hypertension, hypercholesterolemia, hypertrigeleridemia, Diabetes, CVD and psoriatic arthritis. Key results that has emerged out of this study include, a) A significant decrease in patients serum cholesterol, b) Substantial decrease in no of patients with metabolic syndrome, hypercholesterolemia and hypertriglyceridemia, c) Significant decrease in average blood glucose concentration in those patients who is also taking insulin, d) Signiant decrease in PASI, Pain, Nail pittance and DLQI values. The anti-inflammatory action of apremilast administration was supported by a significant decrease in serum inflammatory marker, CRP erythrocyte sedimentation rate. Overall these results indicate that Apremilast can either have a synergistic effect on the ongoing treatment regimen for the co-morbidities. These results highlight, not only the clinical efficacy of apremilast on psoriasis but also on the other co-morbidities warranting publication in this journal but some minor queries are placed here.

Dear Reviewer, thank you for your interest in our manuscript and for your valuable comments aimed at improving its quality.

Comments:

  1. Only serum CRP data is provided for the anti-inflammatory action of apremilast. Could the authors provide supporting data on serum TNFa/IL17 given that TNFa is also a global inflammatory marker involved in other comorbidities while IL17 is directly relavent to psoriasis pathogenesis?

1.Dear reviewer, thank you for the valuable comment. In this study, we did not evaluate IL17 and TNFa. Our observational study focused on routine metabolic values, but we have improved the discussion regarding the role of apremilast as a pleiotropic molecule acting on the IL17 family and TNFα, as they are proinflammatory cytokines involve in both psoriasis and metabolic syndrome. This is a great marker to include in future studies and we appreciate your suggestion.

  1. In Figure 1, even without statin co-treatment there is an apparent reduction in total cholesterol levels. Did the authors perform statistical analysis on this data (since I do not see any n.s. on top of this data)? If the decrease observed is significant, it further strengthen the innate anti-inflammatory activity of apremilast influencing total cholesterol levels.

2.Dear reviewer, thank you for the valuable comment. Statistical significance has not been reported for total cholesterol in this study population, only in patients treated with statins. We have added this information and modified all the graphics to clarify this aspect. Moreover, we have also added information on both LDL and HDL values to better clarify the role of apremilast.

  1. Why authors did not include the total triglyceride, in patients before and after treatment of apremilast, similar to Figure 1 but instead, they provide the no of patients who have recovered from these co-morbidities. As a reviewer/reader, I would be interested to see the extent of triglyceride reduction after apremilast treatment with and without statins.

3.Dear reviewer, thank you for the valuable comment. We have added this information in the results section as Figure 4.

  1. Pairwise statistical analysis need to be included for Figure 6.
  2. Dear reviewer, thank you for the valuable comment. We appreciate your input and agree with your feedback. We have included the pairwise statistical analysis for Figure 6.
  3. Figure 5 legend: I think its T0 not W0.
  4. We agree with your suggestion and have fixed the manuscript accordingly.
  5. Abbreviations such as PASI, NAPSI, VAS, TJC, DLQI need to be expanded at their first use in results. Though this has been expanded in methods, this section comes after the results and discussion.

6.Dear reviewer, thank you for the valuable comment. We have modified the manuscript according to your suggestion.

  1. I do not see the supplementary files online. Have they been uploaded?

7.Dear reviewer, thank you for the valuable comment. We did not upload supplementary file for reviewers but the inform consent for the journal.

8.No details on the ethical approvals obtained for this study is mentioned (is it in the supplementary files?)

8.Dear reviewer, thank you for the comment. The paper adheres to Legislative Decree 81/2008, an Italian legislation addressing occupational health and safety. According to the relevant national legislation, ethical review and approval for this study were waived.

We appreciate all of your feedback and have carefully considered your suggestions to improve our manuscript.

Round 2

Reviewer 1 Report

Comments and Suggestions for Authors

The authors well addressed the issues I pointed out and appropriately revised the article.

Author Response

1.The authors well addressed the issues I pointed out and appropriately revised the article.

  1. Dear Author,thank you for your positive feedback and the interest in our manuscript.

Best regadards,

Elena Campione and Co-authors

Reviewer 2 Report

Comments and Suggestions for Authors

No concerns are remained.

Author Response

1. No concerns are remained.

  1. Dear Author, thank you for your positive feedback and interest in our manuscript.

Best regards,

Elena Campione and Co-authors